# Teaching wiser, Learning smarter: Multi-stage Decoupled Relational Knowledge Distillation with Adaptive Stage Selection

## Abstract

Due to the effectiveness of contrastive-learning-based knowledge distillation methods, there has been a renewed interest on relational knowledge distillation. However, these methods primarily rely on the transfer of angle-wise information between samples, using only the normalized penultimate layer's output as the knowledge base. Our experiments demonstrate that properly harnessing relational information derived from intermediate layers can further improve the effectiveness of distillation. Meanwhile, we found that simply adding distance-wise relational information to contrastive-learning-based methods negatively impacts distillation quality, revealing an implicit contention between angle-wise and distance-wise attributes. Therefore, we propose a **M**ulti-stage **D**ecoupled **R**elational (MDR) knowledge distillation framework equipped with an adaptive stage selection to identify the stages that maximize the efficacy of transferring the relational knowledge. Furthermore, our framework decouples angle-wise and distance-wise information to resolve their conflicts while still preserves complete relational knowledge, thereby resulting in an elevated transferring efficiency and distillation quality. To evaluate the proposed method, we conduct extensive experiments on multiple image benchmarks (*i.e.* CIFAR100, ImageNet and Pascal VOC), covering various tasks (*i.e.* classification, few-shot learning, transfer learning and object detection). Our method exhibits superior performance under diverse scenarios, surpassing the state of the art by an average improvement of 0.88% on CIFAR-100 across extensively utilized teacher-student network pairs.

## 1 Introduction

Over the past decades, unprecedented development in neural-network-based computer vision has created numerous real-world applications, ranging from image classification (He et al., 2016b; Hu et al., 2018; Ma et al., 2018b), object detection (Ren et al., 2015), and semantic segmentation (Long et al., 2015; Zhao et al., 2017). However, these neural networks demand substantial computational and storage resources due to their large model sizes, resulting in expensive and cumbersome model deployment. To address this limitation, various model compression techniques have been systematically explored, such as pruning (He et al., 2017), quantization (Habi et al., 2020), Neural Architecture Search (NAS) (Wan et al., 2020), and Knowledge Distillation (KD) (Hinton et al., 2015). Among them, KD stands out for its compatibility with other compression techniques, superior generalization ability (Zhao et al., 2022; Liu et al., 2021; Chen et al., 2021) and model structure flexibility, thus making it vital in applications such as object detection (Yang et al., 2022; Chong et al., 2022) and Multiple Object Tracking (MOT) (Zhang et al., 2021; Lee et al., 2020).

KD aims to transfer knowledge from a heavy-weight model (teacher) to a light-weight one (student). A straightforward approach is to align the student's output probability distribution with that of the teacher (Hinton et al., 2015). However, due to the limited scope of information in this distribution, subsequent research has shifted towards matching outputs of intermediate layers (Romero et al., 2014; Heo et al., 2019), which further bifurcates into feature-based and relation-based methods. ReviewKD (Chen et al., 2021), a feature-based method, exploits the residual structure to selectively refine the outputs of multiple intermediate layers. In comparison, prominent relation-based approaches excel by combining the relational matrix from multiple samples with contrastive learning

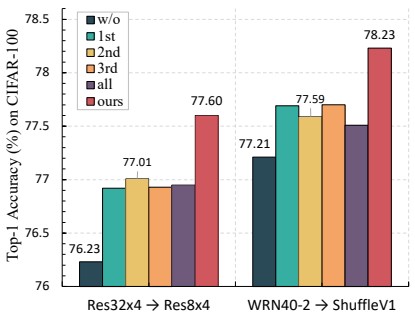 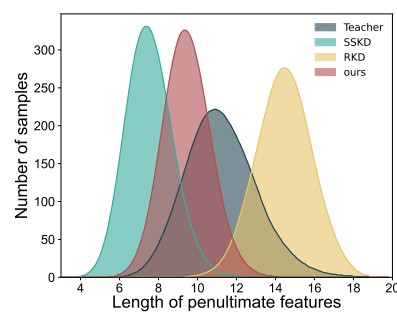

(a) The impact of different stage on Top-1 accuracy of SSKD. MixUp is used in the experiment, and *all* means using all stages' information.

(b) Length distribution of individual sample's penultimate features from models trained by different KD methods (Res32×4 → Res8×4).

Figure 1: Experimental results on relational information of samples in CIFAR100.

in unsupervised domain. Particularly, SSKD (Xu et al., 2020), a seminal work, transfers knowledge by fitting the angle-wise relational matrix composed of positive and negative sample pairs.

Despite these successes, we argue that existing contrastive-learning-based knowledge distillation methods have yet to realize their full potential for two primary reasons. First, these methods only use *single-stage* output features for relationship extraction, thereby overlooking the utility of *multi-stage* relational information between samples. Second, they rely only on inter-sample angle-wise information, neglecting the informative distance component for relational representation.

However, leveraging these missed opportunities presents certain challenges. On one hand, as shown in Fig.1a, the mere incorporation of multi-stage relational information does not necessarily improve the distillation efficacy, even when the volume of transferred information increases. This suggests that raw multi-stage relational data may introduce redundant or even harmful information during the knowledge transfer from the teacher to the student model. On the other hand, Fig. 1b highlights the limitations of solely relying on angle-wise relationships. We calculated the length distribution (denotes the distance from the origin point) of the penultimate layer's output from the student model, in order to eliminate the influence of angle-wise information. Fig. 1b shows the length distribution of penultimate features from models trained by various KD methods, where a larger overlapping area with the teacher's distribution implies greater retention of distance information. This observation reveals that using only the angle-wise relationship between samples for knowledge distillation leads to evident information loss of the length distribution. Moreover, as demonstrated by RKD (Park et al., 2019), directly fitting both angle-wise and distance metrics between samples results in complex, interdependent matrices, and thereby culminates in sub-optimal performance.

To address these constraints, we introduce the Multi-stage Decoupled Relational (MDR) knowledge distillation framework. Utilizing a novel Adaptive Stage Selection (ADSS) strategy, MDR selects the most suitable stages for each sample based on the relational representation capability of both its angle-wise and distance-wise relational information. In addition, the proposed framework decouples inter-sample relationships into angle-wise and length-wise dimensions, allowing for a simultaneous and unconflicted transfer of both types of information. Moreover, in order to prevent Self-supervised Module (SM) from neglecting length information during feature normalization in contrastive-learning-based methods, we present a novel training methodology for SMs that replaces the training paradigm based on contrastive learning with an auxiliary classifier. This modification preserves length attributes while enhancing angle-wise representation capability. Our evaluation demonstrates that MDR framework surpasses the state of the art (SOTA) by an average of 0.88% on CIFAR-100 across extensively utilized network pairs.

To summarize, this paper makes the following contributions:

- We present critical insights into the constraints of the existing contrastive-learning-based knowledge distillation frameworks. We delineate the avenues to further improve the methods through the optimized selection of multi-stage information and the strategic decoupling of angle-wise and length-wise relational representation.

- We propose adaptive stage selection to enable valuable multi-stage information extraction. We further present the concept of relationship decoupling to partition relationships into angle-wise and length-wise components for streamlined training process of the student. In addition, we formulate a new training paradigm for the SM to compensate for the loss of length-wise information and augment its contrastive learning representation capability.

- We cohesively integrate these innovations into a novel MDR distillation framework. Our comprehensive evaluation results show that MDR framework consistently exceeds SOTA performance across extensively utilized network pairs on CIFAR-100, with an accuracy improvement up to 1.22%.

## 2 PRELIMINARY

In this section, we provide a brief review of KD and the details of contrastive-learning-based knowledge distillation methods.

The existing KD methods can be broadly classified into *response-based*, *feature-based*, and *relation-based* methods. In particular, the *response-based* methods transfer the *dark knowledge* from the teacher by approximating the distribution of soft targets, which can be formulated as:

$$\mathcal{L}_{kd} = \tau^2 KL(\sigma(\boldsymbol{z^s}; \tau) \| \sigma(\boldsymbol{z^t}; \tau)), \tag{1}$$

where $\boldsymbol{z^s}$ and $\boldsymbol{z^t}$ are the logits from the student and the teacher respectively; $\sigma(\cdot)$ is the softmax function that produces the category probabilities from the logits, and $\tau$ is a temperature hyper-parameter to scale the smoothness of the distribution; $KL$ means Kullback-Leibler divergence, which is the measurement of dissimilarity between two categorical distributions.

The main idea of the *feature-based* KD methods is to mimic the feature representations between student and teacher, which can be formulated as the following loss function:

$$\mathcal{L}_{feat} = \sum_k \mathcal{L}_f(\mathcal{T}_s(F_k^s), \mathcal{T}_t(F_k^t)), \tag{2}$$

where for stage $k$, $F_k^s$ and $F_k^t$ denote the feature maps from the student and the teacher respectively; $\mathcal{T}_s$, $\mathcal{T}_t$ denote the student and the teacher transformation module respectively; $\mathcal{L}_f(\cdot)$ denotes the function which compute the distance between two feature maps. Using multi-stage information has become the prevailing approach for feature-based methods (Yang et al., 2021; Chen et al., 2021).

In contrast to the methods that distill knowledge from individual samples, the *relation-based* KD methods exploit the relationship between distinct samples, which can be formulated as:

$$\mathcal{L}_{rela}(F_t, F_s) = \mathcal{L}_{R^2}(\psi(t_i, t_j), \psi(s_i, s_j)), \tag{3}$$

where $(t_i, t_j) \in F_t$ and $(s_i, s_j) \in F_s$, $F_t$ and $F_s$ are the sets of feature representations of samples from the teacher and student respectively; $\psi(\cdot)$ denotes the similarity function of $(t_i, t_j)$ or $(s_i, s_j)$; $\mathcal{L}_{R^2}(\cdot)$ is the correlation function of the feature representations between teacher and student (*e.g.*, Huber loss). However, the existing relation-based methods focus on the design of the relational matrix and neglect the valuable multi-stage information.

As the predominant *relation-based* method, contrastive-learning-based knowledge distillation captures inter-sample relationships to transfer knowledge by leveraging the cosine similarity within the representation space. Given a mini-batch containing $N$ samples $\{x_i\}_{i=1:N}$ (*i.e.*, anchor set $\mathcal{P}$), we apply strong data augmentation $t(\cdot)$, such as Random Rotation (Xu et al., 2020) or MixUp (Yu et al., 2022), to each sample and obtain $\{\widetilde{x}_i\}_{i=1:M}$ ((*i.e.*, positive set $\widetilde{\mathcal{P}}$) where $M = 3N$. Both $x_i$ and $\widetilde{x}_i$ are fed into the teacher or student networks to extract their representations $\phi_i = f(x_i), \widetilde{\phi}_i = f(\widetilde{x}_i)$. The similarities between $x_i$ and $\widetilde{x}_i$ can be represented by the following matrix $\mathcal{A}$:

$$\mathcal{A}_{i,j} = cosine(\widetilde{z}_i, z_j) = \frac{dot(\widetilde{z}_i, z_j)}{||\widetilde{z}_i||_2 ||z_j||_2}, \tag{4}$$

where $\widetilde{z}_i$ and $z_j$ are the outputs of SM, which transforms $\widetilde{\phi}_i$ and $\phi_i$ into a contrastive learning representation space. $\mathcal{A}_{i,j}$ represents the similarity between $\widetilde{x}_i$ and $x_j$. $(\widetilde{x}_i, x_i)$ refers to the positive pair and $(\widetilde{x}_i, x_j)_{i \neq j}$ the negative pair. The SM consists of a 2-layer perceptron with a pooling layer,

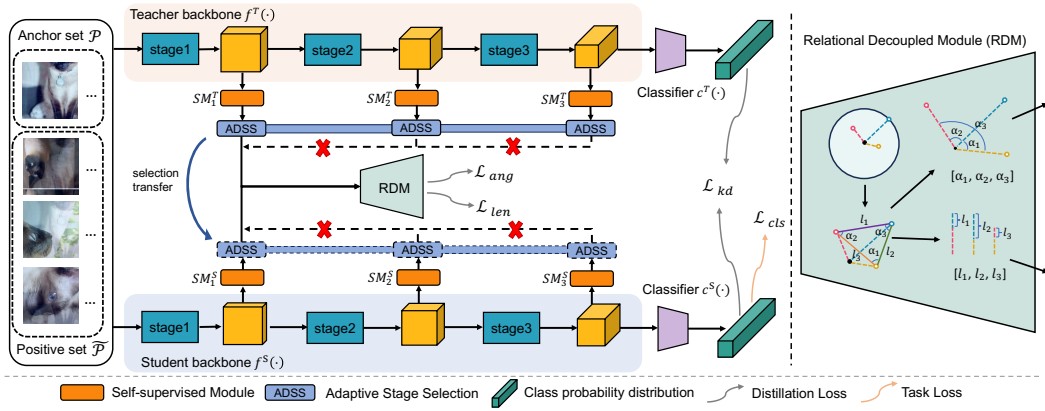

Figure 2: **Illustration of our proposed MDR framework.** ADSS selects the appropriate stage for information transfer (the first stage is selected in this example). Relational Decoupled Module (RDM) transforms the relational information among multiple samples within the corresponding stage into angle-wise (*e.g.* $[\alpha_1, \alpha_2, \alpha_3]$) and length-wise (*e.g.* $[l_1, l_2, l_3]$) representations. $\mathcal{L}_{ang}$ and $\mathcal{L}_{len}$ are used to transfer decoupled relational information. The red cross indicates that the information at this stage is filtered in this case.

which is trained by maximizing the similarity between positive pairs. A commonly used contrastive objective is defined as:

$$\mathcal{L}_{con} = -\sum_i log \frac{\exp(\mathcal{A}_{i,i}/\tau)}{\sum_k \exp(\mathcal{A}_{i,k}/\tau)}. \tag{5}$$

In addition to the angle-wise relational matrix formation, the distance-wise relational matrix between samples can also be used to transfer knowledge (Park et al., 2019), which can be expressed as:

$$\mathcal{D}_{i,j} = ||\widetilde{z}_i - z_j||_2. \tag{6}$$

However, utilizing both the angle-wise and distance-wise matrices simultaneously leads to a degraded performance due to their strongly coupled relationship.

## 3 METHODOLOGY

In light of the aforementioned problems and limitations of the existing methods, we present our proposed framework and featuring an adaptive stage selection strategy followed by the concept of relationship decoupling.

### 3.1 ADAPTIVE STAGE SELECTION STRATEGY

As mentioned in Sec.1, the existing contrastive-learning-based knowledge distillation methods only use single-stage output features for relationship extraction. However, as shown in Fig. 1a, the output of each stage contains valuable angle-wise relational information for the student to learn. To better exploit these information, we adopt a multi-stage framework to transfer knowledge. Specifically, for each distillation stage, both teacher and student networks are equipped with SMs to capture relational information. We augment the data set using MixUp and derive the anchor set $\mathcal{P}$ and the positive set $\widetilde{\mathcal{P}}$. To fully exploit the representation capability, we incorporate both the angle-wise and distance-wise information instead of solely relying on angle-wise relationships in the loss function. Therefore, unlike Eqn. 3, the loss of multi-stage knowledge transfer is represented as:

$$\mathcal{L}_{rela} = \sum_k \sum_{i \in \widetilde{\mathcal{P}}, j \in \mathcal{P}} \mathcal{L}_{R^2}(\mathcal{B}_{i,j}^{s,k} \| \mathcal{B}_{i,j}^{t,k}), \tag{7}$$

where for $k$-th stage, $\mathcal{B}^s$ is a probability matrix, consisting of student's similarity matrix $\mathcal{A}^s$ (Eqn. 4) or $\mathcal{D}^s$ (Eqn. 6) with softmax (with temperature scale $\tau$) along the dimension of all samples from $\mathcal{P}$ in the mini-batch. The same procedure is applied to the teacher to obtain $\mathcal{B}^t$.

As shown in Fig. 1a, incorporating multi-stage relational information straightforwardly does not necessarily improve the distillation efficacy. We argue that every stage contains beneficial information, but using all stages introduce redundant or even harmful information during the knowledge transfer. In order to obtain just enough relational information effectively, we propose an adaptive stage selection strategy to select the most appropriate stage for angle-wise knowledge transfer.

Since both the intra-class (positive pairs) and inter-class (negative pairs) correlation can reflect the representational capability, it is insufficient to use only the cosine similarity of positive pairs as a metric. Moreover, the representation ability at each stages weigh differently towards the final outcome, making it inappropriate to use the absolute value of similarity for direct comparison. Therefore, we use relative numerical ranking instead of absolute cosine similarity. Specifically, we calculate the cosine similarity between each positive sample and all anchor samples using Eqn. 4, and sort them at each stage individually within the mini-batch. This procedure facilitates a holistic comparison of the representational ability for the identical positive samples between different stages. By comparing the order of the similarity between a positive sample and its corresponding anchor in each stage, we select the highest-ranking stage for knowledge transfer.

As illustrated in Fig. 2, we exploit distance-wise relationship, in order to convey more comprehensive information during distillation. As for the stage selection, we use absolute distance as the criterion and also exploit relative numerical ranking by using the following formula:

$$AS(\{M_i^k\}_{k=1}^K) = \arg\min_k \{Rank(M_i^k)\}_{k=1}^K, \tag{8}$$

where $K$ is the number of stages in a network; $M_i^k$ is angle-wise or distance-wise similarity matrix between positive samples $i$ and all anchor samples in the mini-batch; and $Rank$ is the function that sorts the similarity in a descending order.

## 3.2 RELATIONAL DECOUPLED MODULE

As shown in Fig. 1b, exclusively depending on angle-wise relationship during distillation leads to information loss of the length distribution. Therefore, it is crucial to utilize both angle-wise and distance-wise information in order to comprehensively capture the relational information between samples for distillation. Yet, it is challenging to effectively combine these two types of information. For example, RKD directly used two types of information, but the best distillation results are often obtained by taking one of the two. This is because the distance metric contains both angle and length (the latter indicating distance from the origin) information, which may obstruct the comprehension of angle information while learning distance information. In the feature representation space, the sample distance does not align with the principles of contrastive learning as described in Eqn. 5. Specifically, the distance between the positive pairs does not always need to be close. Therefore, conflicts arise when fitting angle-wise and distance-wise relationships simultaneously.

To solve this problem, we propose the concept of relationship decoupling. As illustrated in Fig. 2, we decouple the relationship between samples into angle and length difference, and the latter can be expressed by the following equation:

$$\mathcal{D}iff_{i,j} = \frac{1}{\mu_i} \mid \|\widetilde{z_i}\|_2 - \|z_j\|_2 \mid, \tag{9}$$

where $\mu$ is a normalization factor for length difference. Similar to RKD, we set $\mu$ to be the average length difference between pairs from $\mathcal{P}$ and $\widetilde{\mathcal{P}}$ in the mini-batch:

$$\mu_i = \frac{1}{|\mathcal{P}^2|} \sum_{j \in \mathcal{P}} \mid \|\widetilde{z_i}\|_2 - \|z_j\|_2 \mid. \tag{10}$$

Unlike the traditional way of transferring angle-wise knowledge, we directly use MSE loss to fit the length-wise relational matrix. The length-wise loss and angle-wise loss are defined respectively as:

$$\mathcal{L}_{len} = \sum_{i \in \widetilde{\mathcal{P}}, j \in \mathcal{P}} MSE(\mathcal{D}iff_{i,j}^{s,k}, \mathcal{D}iff_{i,j}^{t,k}) \qquad \text{s.t. } k = AS(\{\mathcal{D}_{i,j}^{t,k}\}_{k=1}^K), \tag{11}$$

$$\mathcal{L}_{ang} = \tau^2 \sum_{i \in \widetilde{\mathcal{P}}, j \in \mathcal{P}} KL(\mathcal{B}_{i,j}^{s,k} \| \mathcal{B}_{i,j}^{t,k}) \qquad \text{s.t. } k = AS(\{\mathcal{B}_{i,j}^{t,k}\}_{k=1}^K). \tag{12}$$

Table 1: Top-1 accuracy (%) comparison of SOTA distillation methods across various teacher-student pairs on CIFAR-100. The numbers in **Bold** and underline denote the best and the second-best results, respectively. For a fair comparison, MixUp and classic KD loss are added to all methods.

| Teacher | WRN40-2 | WRN40-2 | ResNet56 | ResNet110 | VGG13 | ResNet32×4 | ResNet32×4 | ResNet50 | |
|---|---|---|---|---|---|---|---|---|---|
| Acc. | 76.41 | 76.41 | 73.44 | 74.07 | 75.38 | 79.42 | 79.42 | 79.34 | Avg |
| Student | WRN40-1 | WRN16-2 | ResNet20 | ResNet32 | VGG8 | ResNet8×4 | ShuffleV2 | MobileV2 | |
| Acc. | 71.98 | 73.26 | 69.06 | 71.45 | 70.68 | 72.50 | 71.82 | 64.60 | |
| KD | 73.99 | 75.81 | 71.31 | 73.23 | 73.33 | 73.69 | 74.73 | 68.09 | 73.02 |
| FitNet | 74.44 | 75.63 | 71.59 | 73.26 | 74.02 | 75.28 | 75.30 | 66.77 | 73.29 |
| AT | 74.67 | 75.77 | 71.60 | 74.03 | 73.92 | 75.42 | 75.51 | 67.20 | 73.52 |
| SP | 73.91 | 75.44 | 71.02 | 73.88 | 73.31 | 74.09 | 75.20 | 69.11 | 73.25 |
| CC | 73.98 | 75.41 | 71.43 | 74.30 | 73.39 | 74.87 | 75.44 | 69.34 | 73.52 |
| RKD | 73.91 | 75.33 | 70.74 | 73.54 | 73.66 | 74.85 | 75.50 | 68.82 | 73.29 |
| PKT | 74.78 | 75.42 | 71.78 | 73.99 | 73.65 | 74.45 | 76.00 | 68.72 | 73.60 |
| CRD | 74.45 | 75.89 | 71.55 | 74.24 | 74.08 | 75.88 | 76.46 | 69.76 | 74.04 |
| CRCD | 74.41 | 76.07 | 71.49 | 73.92 | 74.31 | 75.50 | 76.23 | 69.99 | 73.99 |
| SSKD | 75.64 | 75.72 | 71.34 | 73.71 | 74.88 | 76.01 | 78.53 | 71.91 | 74.72 |
| ReviewKD | 75.41 | 76.42 | 72.04 | 74.10 | 75.03 | 75.91 | 78.02 | 70.21 | 74.64 |
| DKD | 75.02 | 76.44 | 72.09 | 74.39 | 74.91 | 76.49 | 76.58 | 70.51 | 74.56 |
| ML-LD | 74.89 | 76.45 | 71.64 | 73.85 | 74.68 | 75.60 | 76.88 | 70.79 | 74.35 |
| **Ours** | **76.79** | **77.09** | **72.77** | **75.18** | **75.97** | **77.94** | **79.27** | **72.52** | **75.94** |

The final loss for the student network is the combination of aforementioned terms, including the original training loss $\mathcal{L}_{cls}$, the response-based loss $\mathcal{L}_{kd}$, and the relation-based loss $\mathcal{L}_{ang}$ and $\mathcal{L}_{len}$:

$$\mathcal{L} = \lambda_1 \mathcal{L}_{cls} + \lambda_2 \mathcal{L}_{kd} + \lambda_3 \mathcal{L}_{ang} + \lambda_4 \mathcal{L}_{len}, \tag{13}$$

where the $\lambda_i$ is the balancing weight.

Before training the student, we freeze the teacher's backbone and train the SM. SM is typically trained by explicitly improving the representational ability of contrastive learning (Eqn. 5) in existing methods, which neglect length-wise information during feature normalization to prioritize angle-wise relationships. To preserve length-wise information while maintaining the representational ability of contrastive learning, we place a classifier behind each SM and directly use cross-entropy (CE) loss. This approach ensures that the dimensions of the outputs are consistent while retaining the relational information. Compared with the prior training methods, the contrastive learning representational ability of SM is further amplified with CE loss. Moreover, training through a classifier is more effective for SM to obtain the global information of the data set, rather than the relational information between samples within a mini-batch.

## 4 EXPERIMENTS

To demonstrate the effectiveness of our work, we evaluate MDR in various tasks: *classification*, *few-shot learning*, *transfer learning* and *object detection*. Moreover, we present various ablation study for the proposed method. Besides, our codes will be publicly available for reproducibility.

### 4.1 EXPERIMENTAL SETTINGS

We conduct evaluations on standard CIFAR-100 (Krizhevsky et al., 2009) and ImageNet (Russakovsky et al., 2015) benchmarks across the widely applied network families including ResNet (He et al., 2016a), WRN (Zagoruyko & Komodakis, 2016a), VGG (Simonyan & Zisserman, 2015), MobileNet (Sandler et al., 2018), ShuffleNet (Ma et al., 2018a) (see Appendix A.1 for the details of these datasets and related evaluation metrics). Moreover, we employ the SIL-10 (Coates et al., 2011) and TinyImagenet (Russakovsky et al., 2015) datasets to assess the transferability of learned representations generated by distillation method. We compare MDR with a wide range of representative KD methods, including KD (Hinton et al., 2015), FitNets (Romero et al., 2014), AT (Zagoruyko & Komodakis, 2016b), SP (Tung & Mori, 2019), CC (Peng et al., 2019), RKD (Park et al., 2019), PKT (Passalis & Tefas, 2018), OFD (Heo et al., 2019), CRD (Tian et al., 2019), SSKD (Xu et al., 2020), CRCD (Zhu et al., 2021), ReviewKD (Chen et al., 2021), DKD (Zhao et al., 2022), ML-LD (Jin et al., 2023). Due to the page limitation, more settings can be found in Appendix A.2.

Table 2: Top-1 and Top-5 accuracy (%) comparisons of SOTA distillation methods on ImageNet. Part of the compared results are from (Jin et al., 2023). * means the results from our reproduction.

| Acc. | Teacher | Student | KD | AT | RKD | CRD | SSKD* | ReviewKD | DKD | ML-LD* | Ours |
|---|---|---|---|---|---|---|---|---|---|---|---|
| Top-1 | 73.31 | 69.75 | 70.66 | 70.70 | 71.34 | 71.38 | 71.41 | 71.61 | _71.70_ | 71.28 | **71.92** |
| Top-5 | 91.42 | 89.07 | 89.88 | 90.00 | 90.37 | 90.49 | 90.44 | 90.51 | _90.51_ | 90.15 | **90.68** |

Table 3: Linear classification accuracy (%) of transfer learning on the student ResNet8×4 pre-trained using the teacher ResNet32×4.

| Transferred Dataset | Baseline | KD | FitNet | RKD | CRD | SSKD | ReviewKD | DKD | Ours |
|---|---|---|---|---|---|---|---|---|---|
| CIFAR100→SIL-10 | 69.76 | 69.56 | 70.94 | 71.41 | 70.76 | 71.89 | 71.90 | _72.15_ | **73.56** |
| CIFAR100→TinyImageNet | 34.29 | 34.77 | 38.07 | 38.02 | 38.17 | 38.56 | 38.54 | _38.74_ | **39.88** |

## 4.2 COMPARISON WITH THE STATE OF THE ARTS

**Results on CIFAR-100.** We compare our MDR with representative distillation methods using a variety of teacher-student pairs, with both identical and different architectural styles on the CIFAR-100 dataset. As shown in Table 1, our MDR consistently outperforms other methods by a significant margin. Specifically, our method achieves an average of 0.88% accuracy improvement over the current top-performing methods for both identical and distinct architecture pairs. The amount of accuracy improvement is larger than those of many previous methods. Moreover, there's an average improvement of 1.22% in accuracy over SSKD (more experiments in Appendix A.3). These results indicate that our proposed MDR can effectively exploit the decoupled relationship across multiple stages between samples for knowledge distillation. Note that the student's accuracy surpasses the teacher's in certain identical architecture pairs, such as WRN40-2→WRN40-1. This underscores our method's capability to comprehensively extract more valuable information from teachers.

**Results on ImageNet.** We further evaluated a teacher-student pair on the large-scale ImageNet, using ResNet34 as a teacher and ResNet18 as a student. As shown in Table 2, our MDR achieves the best performance in both Top-1 and Top-5 accuracy categories. Specifically, MDR improves the accuracy by 0.51% over SSKD for Top-1 accuracy. These ImageNet results highlight the effectiveness of MDR on large-scale datasets.

**Transferability of Learned Representations.** Beyond achieving superior accuracy on the object dataset, it is imperative for the student network to produce generalized feature representations that can exhibit robust transferability to novel semantic recognition datasets. To this end, we adopt the strategy of freezing the backbone $f^S(\cdot)$ that has been pre-trained on the upstream CIFAR-100. We then train two linear classifiers based on the fixed penultimate features for downstream classification on the STL-10 and TinyImageNet, respectively (Tian et al., 2019). Table 3 shows the ability of transfer learning using different KD methods. Specifically, our MDR method outperforms the best-competing DKD by an accuracy gain of 1.41% on STL-10 and an accuracy gain of 1.14% on TinyImageNet, demonstrating its superior transferability to various recognition tasks.

**Efficiency under Few-shot Scenario.** We evaluate our method against conventional KD, CRD, SSKD and PACKD (Yu et al., 2022) in a few-shot learning environment, using retention rates of 25%, 50%, and 75% of the original training samples. To ensure a fair comparison, we maintain a consistent data split strategy for each few-shot scenario, while keeping the original test set intact. Our evaluation utilizes the ResNet56-ResNet20 pair. As depicted in Table 4, our method consistently outperforms the other techniques by large margins across various few-shot scenarios. Notably, compared with the baseline trained on the complete set, our method achieves higher accuracy with only 25% of the training data.

Table 4: Top-1 accuracy (%) comparison on CIFAR-100 under few-shot scenario with various percentages of training samples.

| Percentage | KD | CRD | SSKD | PACKD | Ours |
|---|---|---|---|---|---|
| 25% | 64.40 | 64.71 | 67.82 | _68.63_ | **69.11** |
| 50% | 68.37 | 68.90 | 70.08 | _70.73_ | **71.17** |
| 75% | 69.97 | 70.86 | 70.47 | _71.70_ | **72.35** |

This outcome is attributed to our method's ability to effectively learn general relational information from limited data. In comparison, the previous methods typically focus on mimicking inductive biases from intermediate feature maps or incomplete relationships, which may overfit on the limited dataset and reduce generalization on the test set.

**Transferability for Object Detection.**  We further evaluate the student network ResNet-18, which is pre-trained with the teacher ResNet-34 on ImageNet, as a backbone for downstream object detection on Pascal VOC (Everingham et al., 2010). For this evaluation, we adopt the Faster-RCNN (Ren et al., 2017) framework, adhering to the standard data pre-processing and fine-tuning protocols. Table 5 shows our method's superior detection performance, surpassing the original baseline by 2.24% mAP and the best-competing SSKD method

Table 5: Comparison of detection mAP (%) on Pascal VOC using ResNet-18 as the backbone pre-trained by various KD methods.

| Baseline | KD | CRD | SSKD | Ours |
|---|---|---|---|---|
| 76.18 | 77.06 | 77.36 | 77.60 | **78.42** |

by 0.82% mAP. These results underscore our method's efficacy in guiding a network to achieve superior feature representations for diverse semantic recognition tasks.

### 4.3 ABLATION STUDIES

In this section, we provide ablation studies to analyze the effects of each component of MDR. The experiments are conducted on CIFAR-100 for classification task.

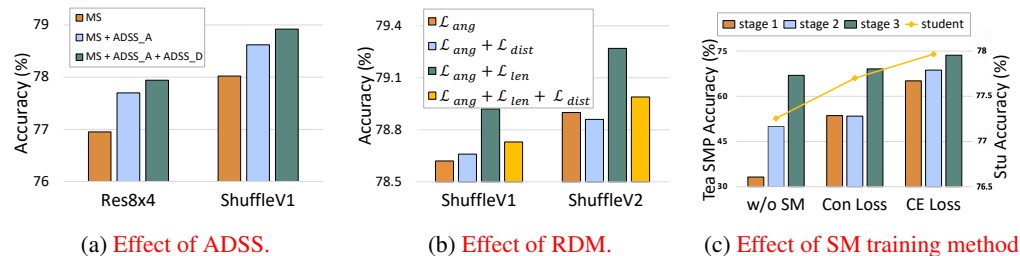

(a) Effect of ADSS.   (b) Effect of RDM.   (c) Effect of SM training method.

Figure 3: Ablation study on CIFAR100. Student network ResNet8×4, ShuffleV1 and ShuffleV2, are trained under teacher network ResNet32×4.

**Effect of Adaptive Stage Selection.**  As shown in Fig. 3a, MS means using multi-stage decoupled relational information to transfer, which contains angle-wise and length-wise information in each stage. Applying angle-wise adaptive stage selection strategy (MS + ADSS_A) substantially boosts the accuracy upon the original multi-stage information, indicating that we extract a larger amount of beneficial angle-wise relationship. As we further add distance-wise adaptive stage selection strategy (MS + ADSS_A + ADSS_D), an even higher accuracy is achieved thanks to the positive contribution of valuable distance-wise information.

**Effect of Relational Decoupled Module.**  To explore the effectiveness of proposed RDM, we conduct the evaluation in three variants: only using angle-wise information ($\mathcal{L}_{ang}$), angle-wise and distance-wise information ($\mathcal{L}_{ang} + \mathcal{L}_{dist}$), both angle-wise and length-wise information ($\mathcal{L}_{ang} + \mathcal{L}_{len}$) and all three information ($\mathcal{L}_{ang} + \mathcal{L}_{len} + \mathcal{L}_{dist}$). The results are shown in Fig. 3b, where coupled information ($\mathcal{L}_{dist}$) often have a negative impact on distillation, and RDM boosts the accuracy compared to the others.

To assess the impact of CE loss for SM training, we compare the teacher's SMP accuracy and student's accuracy under three cases: no SM, with contrastive loss, and with CE loss. SMP accuracy shows the faction of positive samples correctly assigned to the corresponding anchor. As shown in Fig. 3c, compared with the no-SM case, training with contrastive loss improves SMP accuracy, which is further improved by CE loss in both types of accuracy.

**The number of adaptive stages.**  We validate various number of adaptive stages based on angle and distance respectively: 1/2/3/4 with two teacher-student pairs, including identical and distinct architectures. For the fairness of the comparison, when verifying the number of adaptive stages based on

Table 6: Ablations on the number of adaptive stages.

| Tea-Stu pair | info category | N = 1 | N = 2 | N = 3 | N = 4 |
|---|---|---|---|---|---|
| VGG13→VGG8 | angle-wise | **75.97** | 75.88 | 75.83 | 75.53 |
| VGG13→VGG8 | length-wise | **75.97** | 75.81 | 75.84 | 75.66 |
| Res50→MobileV2 | angle-wise | **72.52** | 72.32 | 72.19 | 72.01 |
| Res50→MobileV2 | length-wise | **72.52** | 72.29 | 72.33 | 72.21 |

angle, length-wise one is fixed to be 1, and vice versa. As shown in Table 6, regardless of angle-wise or length-wise information, the best result is achieved when the number of adaptive stages is 1. Combined with stage selection, the accuracy improves steadily.

Due to the page limitation, more ablation studies and experiment analysis can be found in Appendix A.4 and A.5, and more related visualizations can be found in Appendix A.7.

# 5 RELATED WORK

Knowledge distillation trains a smaller network using the knowledge from a larger network. Based on the types of the distilled knowledge, existing KD frameworks can be divided into three categories: *response-based*, *feature-based* and *relation-based* methods (Gou et al., 2021).

***Response-based*** KD, also known as the classic KD (Hinton et al., 2015), usually relies on the neural response of the last output layer of the teacher model. The main idea is to directly mimic the final prediction (logits) of the teacher model. DKD (Zhao et al., 2022) proposes a decoupled approach using the fundamental concept of KD. Unlike our proposed decoupled approach, DKD only separates the output categories into target and non-target classes, and assigns different importance to them. HSAKD (Yang et al., 2021) trains separate classifiers for each stage while transferring multi-stage response-based information.

***Featured-based*** KD, represented by FitNet (Romero et al., 2014), encourages the student models to mimic the intermediate-level features from the hidden layers of teacher models. VID (Ahn et al., 2019) and PKT (Passalis & Tefas, 2018) reformulate knowledge distillation as a procedure of maximizing the mutual information between the teacher and the student networks. There are also other methods using multi-stage information to transfer knowledge. OFD (Heo et al., 2019) uses a novel distance function to transfer the information from the teacher to the student; ReviewKD (Chen et al., 2021) proposes a new multi-stage architecture that allows the student to select the appropriate teacher stage for distillation. In contrast, our method employs an adaptive stage selection strategy to extract the most relevant relational information applicable to distillation for different samples.

***Relation-based*** KD emphasizes the exploitation of relationships between distinct layers or samples. FSP (Yim et al., 2017) guides the student model by generating a relation matrix between different layers of the teacher model. SP (Tung & Mori, 2019), CC (Peng et al., 2019), and RKD (Park et al., 2019) utilize the relationships between samples to guide the student in learning higher-dimensional representations. Leveraging the success of contrastive learning in unsupervised tasks (Peng et al., 2019; Park et al., 2019), many methods utilize the representation space of contrastive learning to model the relationships between samples. CRD (Tian et al., 2019) pioneered the integration of contrastive learning into knowledge distillation, SSKD (Xu et al., 2020) separately trains the teacher's SM to extract richer knowledge, PACKD (Yu et al., 2022) uses an optimal transport-based positive pair similarity weighting strategy to better transfer discriminative information from teachers to students. However, all of the existing contrastive-learning-based methods extract relationships at the penultimate feature layer. According to our experiments, we found that effective relational information can also be extracted from intermediate layers. Therefore, we propose a multi-stage distillation framework with adaptive stage selection strategy to comprehensively extract relational knowledge between samples. Moreover, our novel method decouples the relationship between samples into angle and length difference, compensating information loss in length-wise relationships in the existing contrastive-learning-based methods.

# 6 CONCLUSION

In this paper, we propose a novel framework equipped with an adaptive stage selection strategy for relation-based knowledge distillation, which enables efficient extraction of relational information across multiple stages. By decoupling the relationship into angle and length difference and introducing a novel training method for the self-supervised module, our approach enables the student to acquire knowledge more effectively. Experiment results show that our method significantly surpasses SOTA performance on the standard image classification benchmarks in the field of KD. It also opens the door for further improvements of knowledge transfer methods based on relationship.

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

# A   APPENDIX

## A.1   DATASETS AND METRICS

**CIFAR-100.** CIFAR-100 (Krizhevsky et al., 2009) contains 50K images for training and 10K images for testing, labeled into 100 fine-grained categories. The size of each image is 32×32. We evaluate the proposed MDR on this dataset with image recognition and report the top-1 accuracy.

**ImageNet.** ImageNet (Russakovsky et al., 2015) consists of 1.2M images for training and 50K images for validation, covering 1,000 categories. All images are resized to $224 \times 224$ during training and testing. We report the top-1 and top-5 accuracy on this dataset for image recognition.

**STL-10 and TinyImageNet.** STL-10 (Coates et al., 2011) is composed of 5K labeled training images and 8K test images in 10 classes. TinyImageNet (Russakovsky et al., 2015) is composed of 100K training images and 10k test images in 200 classes. We evaluate the proposed MDR on this dataset with image recognition and report the top-1 accuracy.

**Pascal VOC.** Following the consistent protocol, we use Pascal VOC (Everingham et al., 2010) trainval07 + 12 for training and test07 for evaluation. The result set consists of 16551 training images and 4952 test images in 20 classes. The image scale is $1000 \times 600$ pixels during training and inference. The comparison of detection performance toward average precision (AP) on individual classes and mean AP (mAP).

## A.2   IMPLEMENTATIONS

**Training details.** On CIFAR-100, the batch size and initial learning rate are set to 64 and 0.05. We train the models for 240 epochs in total with SGD optimizer, and decay the learning rate by 0.1 at 150, 180, and 210 epochs. The weight decay and the momentum are set to 5e-4 and 0.9. We set $\tau$ in $\mathcal{L}_{kd}$ for $\mathcal{P}$ to be 4, $\widetilde{\mathcal{P}}$ to be 0.75, $\tau$ in $\mathcal{L}_{ang}$ and $\mathcal{L}_{len}$ to be 0.5. We set $\lambda_1 = 0.1$, $\lambda_2 = 2.7$, $\lambda_3 = 300$, $\lambda_4 = 1.0$ in Eqn. 13. We conduct experiments on one Tesla T4 GPU. On ImageNet, we adopt the SGD optimizer to train the student networks for 100 epochs with a batch size of 512. The initial learning rate is 0.2 and decayed by 10 when the epoch is 30, 60 and 90. The optimizer with 5e-4 weight decay and 0.9 momentum is adopted. We set $\tau$ in $\mathcal{L}_{kd}$ for $\mathcal{P}$ to be 1, $\widetilde{\mathcal{P}}$ to be 1, $\tau$ in $\mathcal{L}_{ang}$ and $\mathcal{L}_{len}$ to be 0.5. We set $\lambda_1 = 1.0$, $\lambda_2 = 2.0$, $\lambda_3 = 300$, $\lambda_4 = 1.0$ in Eqn. 13. Our inplementation on Pascal VOC for object detection follows the same setting used in (Yang et al., 2021). Following the consistent protocol, we use trainval07 + 12 for training and test07 for evaluation. We train the model by SGD optimizer with a momentum of 0.9 and a weight decay of $1 \times 10^{-4}$. The initial learning rate starts at 0.01 and is decayed by a factor of 10 at the third epoch within a total of four epochs. We conduct experiments on 4 Tesla V100 GPUs. On STL-10 and TinyImageNet, we train the linear classifiers by the SGD optimizer with a momentum of 0.9, a batch size of 64 and a weight decay of 0. The initial learning rate starts at 0.1 and is decayed by a factor of 10 at 30, 60 and 90 epochs within the total 100 epochs. We conduct experiments on one Tesla T4 GPU.

To ensure a fair comparison, we use the same data augmentation MixUp and classic KD loss for all methods. For the identical network, we maintain the same training hyperparameters, including learning rate, epoch and weight dacay. In particular, we align the data augmentation methods and hyperparameter configurations used in the ML-LD original code with those in our other experiments.

During the training stage of teacher's SMs, we connect a classifier after each SM (composed of a layer of fully connected), and directly uses category information for supervised learning. In this process, except for SM and classifier, the backbone part of the network remains frozen. On CIFAR100, we train the SMs for a total of 60 epochs, with the learning rate decayed by a factor of 10 at the 30th and 45th epochs. On ImageNet, we train the SMs for 30 epochs, with the learning rate decayed by 10 at the 10th and 20th epoch. Other settings remain consistent with the distillation process.

All experiments are conducted based on the Pytorch framework.

**Architectural Design of SMs.** As discussed in the main paper, we attach one SM after each convolutional stage. The SM is composed of global average pooling(GAP) and two fully-connected(FC) layer. For training teacher SMs, we attach one FC layer for CE loss, where the input dimension is

Table 7: Architectural details of SMs for various networks for CIFAR-100 classification.

| Network Name | $SM_1GAP$ | $SM_2GAP$ | $SM_3GAP$ | $SM_4GAP$ | SS Modules input dim | Classifier |
|---|---|---|---|---|---|---|
| WRN-40-2 | $(8 \times 8)$ | $(8 \times 8)$ | $(8 \times 8)$ | - | [512, 256, 128, -] | (128, 100) |
| WRN-40-1 | $(8 \times 8)$ | $(8 \times 8)$ | $(8 \times 8)$ | - | [256, 128, 64, -] | (128, 100) |
| WRN-16-2 | $(8 \times 8)$ | $(8 \times 8)$ | $(8 \times 8)$ | - | [512, 256, 128, -] | (128, 100) |
| resnet56 | $(8 \times 8)$ | $(8 \times 8)$ | $(8 \times 8)$ | - | [256, 128, 64, -] | (128, 100) |
| resnet20 | $(8 \times 8)$ | $(8 \times 8)$ | $(8 \times 8)$ | - | [256, 128, 64, -] | (128, 100) |
| ResNet32×4 | $(8 \times 8)$ | $(8 \times 8)$ | $(8 \times 8)$ | - | [1024, 512, 256, -] | (128, 100) |
| ResNet8×4 | $(8 \times 8)$ | $(8 \times 8)$ | $(8 \times 8)$ | - | [1024, 512, 256, -] | (128, 100) |
| ResNet-50 | $(8 \times 8)$ | $(8 \times 8)$ | $(4 \times 4)$ | $(2 \times 2)$ | [1024, 512, 1024, 2048] | (128, 100) |
| VGG-13 | $(16 \times 16)$ | $(8 \times 8)$ | $(4 \times 4)$ | $(4 \times 4)$ | [128, 256, 512, 512] | (128, 100) |
| VGG-8 | $(16 \times 16)$ | $(8 \times 8)$ | $(4 \times 4)$ | $(4 \times 4)$ | [128, 256, 512, 512] | (128, 100) |
| MobileNetV2 | $(8 \times 8)$ | $(8 \times 8)$ | $(4 \times 4)$ | $(2 \times 2)$ | [48, 16, 48, 160] | (128, 100) |
| ShuffleNetV1 | $(8 \times 8)$ | $(8 \times 8)$ | $(4 \times 4)$ | - | [960, 480, 960, -] | (128, 100) |
| ShuffleNetV2 | $(8 \times 8)$ | $(8 \times 8)$ | $(4 \times 4)$ | - | [464, 232, 464, -] | (128, 100) |

Table 8: Architectural details of SMs for various networks for ImageNet classification.

| Network Name | $SM_1GAP$ | $SM_2GAP$ | $SM_3GAP$ | $SM_4GAP$ | SM's input dim | Classifier |
|---|---|---|---|---|---|---|
| ResNet-34 | - | $(28 \times 28)$ | $(14 \times 14)$ | $(7 \times 7)$ | [-, 128, 256, 512] | (256, 1000) |
| ResNet-18 | - | $(28 \times 28)$ | $(14 \times 14)$ | $(7 \times 7)$ | [-, 128, 256, 512] | (256, 1000) |
| ResNet-50 | - | $(28 \times 28)$ | $(14 \times 14)$ | $(7 \times 7)$ | [-, 512, 1024, 1024] | (256, 1000) |
| MobileNetV1 | - | $(14 \times 14)$ | $(7 \times 7)$ | $(1 \times 1)$ | [-, 512, 1024, 2048] | (256, 1000) |

same as the dimension of SM's output feature (*e.g.*, 128 on CIFAR-100, 256 on ImageNet) and the output dimension is same as the number of categories.

We illustrate the overall architecture of SMs for various networks on CIFAR-100 on Table 7 and ImageNet on Table 8, including the family of WRN, ResNet, VGG, MobileNet and ShuffleNet. In the specific case of ResNet34→ResNet18 in the ImageNet classification task, we specifically utilize the output of the second to fourth stages for extracting relational information. Our decision is grounded on the observation that the accuracy of SMP in the first stage is less than 10%. This finding implies that the first stage lacks significant representation information necessary for effective contrastive learning.

### A.3    CIFAR-100 AND IMAGENET RESULTS ON OTHER TEACHER-STUDENT PAIRS.

Table 9: Top-1 accuracy (%) comparison of SOTA distillation methods across various teacher-student pairs on CIFAR-100 (as a supplement to Table 1). The numbers in **Bold** and underline denote the best and the second-best results, respectively.

| Teacher-Student pair | KD | FitNet | AT | RKD | CRD | DKD | ML-LD | SSKD | ReviewKD | Ours |
|---|---|---|---|---|---|---|---|---|---|---|
| ResNet32×4→ShuffleV1 | 74.52 | 73.76 | 76.37 | 74.00 | 75.66 | 76.68 | 76.81 | 78.37 | 77.60 | **78.92** |
| ResNet50→VGG8 | 73.51 | 73.29 | 73.88 | 73.84 | 74.10 | 75.46 | 75.54 | 76.02 | 75.41 | **76.52** |
| WRN40-2→ShuffleV1 | 75.55 | 76.19 | 77.03 | 75.71 | 77.22 | 76.91 | 76.52 | 77.21 | 77.39 | **78.47** |

To complement the data presented in Table. 1, we conducted additional experiments exploring different network configurations. We observed that, in comparison to network pairs with the same architecture, the distillation improvement achieved by using different network architectures was relatively minor. Furthermore, our distillation method consistently enhanced the accuracy of the student model to a level surpassing that of the teacher model. This outcome validates the effectiveness of knowledge extraction from the teacher model and its successful transfer to the students.

We further evaluated a teacher-student pair on the large-scale ImageNet, using ResNet50 as a teacher and MobileNetV1 as a student. As shown in Table 10, our MDR achieves the best performance in

Table 10: Top-1 and Top-5 accuracy (%) comparisons of SOTA distillation methods on ImageNet. Part of the compared results are from (Zhao et al., 2022) and (Jin et al., 2023). * means the result of our reproduction.

| Acc. | Teacher | Student | KD | AT | OFD | CRD | SSKD* | ReviewKD | DKD | ML-LD* | Ours |
|---|---|---|---|---|---|---|---|---|---|---|---|
| Top-1 | 76.16 | 68.87 | 68.58 | 69.56 | 71.25 | 71.37 | 72.48 | 72.56 | 72.05 | 72.04 | **72.84** |
| Top-5 | 92.86 | 88.76 | 88.98 | 89.33 | 90.34 | 90.41 | 91.17 | 91.00 | 91.05 | 91.10 | **91.19** |

both Top-1 and Top-5 accuracy categories. Specifically, MDR improves the accuracy by 0.36% over SSKD for Top-1 accuracy. These ImageNet results highlight the effectiveness of MDR on large-scale datasets.

### A.4 ABLATION STUDIES.

**ADSS with different distance-wise criterion.** Table 6 summarizes the effects of different distance-wise criterion. To ensure a fair comparison, different experiments are carried out under the same angle-wise adaptive stage selection strategy.

In addition to the baseline of not using distance information(w/o L), we take the All Stages' outputs (AS) and the Penultimate Layer's outputs (PL), introduced by RKD. We also consider relative longest length (RLL), which means selecting the stage where the length order of the sample is the highest, and the opposite for relative shortest length (RSL). Additionally, we exploit the correct class confidence for auxiliary classifier's output (CS) to determine the best stage. Furthermore, we believe that although distance and angle are decoupled in learning, angle-wise information needs to be referred to in the judgment of the most appropriate stage of distance, so the anglular information between samples is also integrated into the judgment: directly using the Decision of angle-wise adaptive stage selection strategy (DA) and use of Relative Shortest Distance (RSD).

Table 11: Ablations on distance-wise criterion.

| Teacher-Student | w/o L | AS | PL | RLL | RSL | CS | DA | RSD |
|---|---|---|---|---|---|---|---|---|
| WRN40-2→WRN40-1 | 76.47 | 76.43 | 76.40 | 76.59 | 76.44 | 76.55 | 76.68 | **76.79** |
| WRN40-2→WRN16-2 | 76.81 | 76.76 | 76.80 | 76.90 | 76.81 | 76.79 | 76.88 | **77.09** |
| Res56→Res20 | 72.34 | 72.33 | 72.31 | 72.30 | 72.39 | 72.55 | 72.42 | **72.77** |
| Res110→Res32 | 74.86 | 74.70 | 74.99 | 75.10 | 75.01 | 74.89 | 74.87 | **75.18** |
| VGG13→VGG8 | 75.69 | 75.66 | 75.70 | 75.61 | **75.99** | 75.74 | 75.67 | 75.97 |
| ResNet32×4→ResNet8×4 | 77.60 | 77.65 | 77.69 | 77.75 | 77.73 | 77.50 | 77.59 | **77.94** |
| ResNet32×4→ShuffleV2 | 78.89 | 78.80 | 79.05 | 79.09 | 78.88 | 79.02 | 79.01 | **79.27** |
| ResNet50→MobileV2 | 72.37 | 72.21 | 72.31 | 72.41 | **72.55** | 72.20 | 72.30 | 72.52 |
| ResNet32×4→ShuffleV1 | 78.61 | 78.77 | 78.71 | 78.66 | 78.57 | 78.90 | 78.44 | **78.92** |
| ResNet50→VGG8 | 76.33 | 76.23 | 76.31 | 76.44 | 76.31 | 76.33 | 76.33 | **76.52** |
| WRN40-2→ShuffleV1 | 78.23 | 78.29 | 78.26 | 78.31 | 78.44 | 78.21 | 78.38 | **78.47** |

Table. 11 shows that utilizing the RSD as the criterion for distance-wise adaptive stage selection yields the most effective performance overall. Apart from RSD, CS exhibits a positive influence on the majority of network pairs. Notably, RSL has attained superior outcomes on VGG13→VGG8 and ResNet50→MobileV2, albeit with greater instability compared to RSD. The other criterion demonstrate considerably smaller positive effects than RSD. Therefore, we ultimately adopt RSD as the criterion for distance-wise adaptive stage selection.

**Effect of Adaptive Stage Selection.** We extended the experiment in Fig. 3a. On different teacher-student pairs, we experimented by adding only multi-stage angle-wise information (MS_A), multi-stage angle-wise and length-wise information (MS), and adding ADSS strategy only on angle-wise information (MS + ADSS_A), and using ADSS strategy for angle-wise and length-wise information at the same time (MS + ADSS_A + ADSS_D).

Table 12: Ablations on Adaptive Stage Selection.

| Teacher-Student | MS_A | MS | MS+ADSS_A | MS+ADSS_A+ADSS_D |
|---|---|---|---|---|
| WRN40-2→WRN40-1 | 75.97 | 76.10 | 76.68 | **76.79** |
| WRN40-2→WRN16-2 | 76.23 | 76.40 | 76.92 | **77.09** |
| Res56→Res20 | 71.66 | 71.89 | 72.50 | **72.77** |
| Res110→Res32 | 74.08 | 74.31 | 74.92 | **75.18** |
| VGG13→VGG8 | 75.13 | 75.21 | 75.66 | **75.97** |
| ResNet32×4→ResNet8×4 | 76.67 | 76.95 | 77.69 | **77.94** |
| ResNet32×4→ShuffleV2 | 78.61 | 78.69 | 79.11 | **79.27** |
| ResNet50→MobileV2 | 71.88 | 71.93 | 72.21 | **72.52** |
| ResNet32×4→ShuffleV1 | 78.45 | 78.51 | 78.84 | **78.92** |
| ResNet50→VGG8 | 76.09 | 76.10 | 76.44 | **76.52** |
| WRN40-2→ShuffleV1 | 77.48 | 77.63 | 78.20 | **78.47** |

**Statistics on the number of samples in each stage with ADSS.**

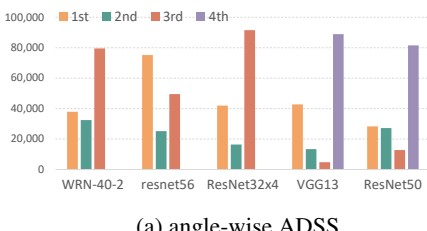

(a) angle-wise ADSS.

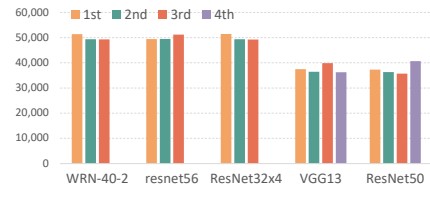

(b) distance-wise ADSS.

Figure 4: Statistics on the number of samples in each stage with ADSS on CIFAR-100.

We performed a statistical analysis of the number of stage selections based on angle and distance for various teacher networks in the CIFAR-100 training set. The results are illustrated in Fig. 4.

As shown in Fig. 4a, regarding the angle-wise ADSS, notable disparities exist in the number of choices across different stages. However, a consistent trend is observed: the proportion of choices is generally higher in the initial and final stages, while relatively lower in the intermediate stages. A plausible explanation is that in the initial stage, the feature extraction capability is limited, primarily capturing shallow features of the original input. As the network deepens, subsequent stages exhibit stronger feature extraction abilities, with a heightened focus on local characteristics. The final stage, which connects to the classifier, is essential to discern significant feature difference among distinct categories, resulting in the strongest representation capability.

By employing the distance-wise ADSS, we observed a relatively small variation in the number of samples across each stage. This finding suggests that the distance-wise representation space exhibits weaker correlation with the network depth, as compared to the angle-wise one.

**Sensitivity analysis for angle-wise and length-wise loss.**

Table 13: Ablations on angle-wise and length-wise weight.

| $\lambda_3$ | 50 | 100 | 300 | 500 | 700 | 900 |
|---|---|---|---|---|---|---|
| Top-1 | 77.71 / 72.40 | 77.83 / **72.56** | **77.94** / 72.52 | 77.81 / 72.45 | 77.80 / 72.33 | 77.66 / 72.23 |
| $\lambda_4$ | 0.2 | 0.5 | 1.0 | 1.5 | 2.0 | 4.0 |
| Top-1 | 77.69 / 72.33 | 77.85 / 72.46 | **77.94 / 72.52** | 77.80 / 72.40 | 77.71 / 72.33 | 77.60 / 72.21 |

We performed a sensitivity analysis for angle-wise and length-wise loss for various teacher-student networks in the CIFAR-100. ResNet32×4→ResNet8×4 (left) and ResNet50→MobileNetV2 (right) are set as the teacher and the student, respectively. As shown in Table. 13, the best results were achieved when $\lambda_3$ is around 300 and $\lambda_4$ is 1.0. Our other experiments were conducted under this set of hyperparameter settings.

## A.5 EXPERIMENT ANALYSIS.

Table 14: Comparison of detection AP (%) on individual classes and mAP (%) on PASCAL VOC using ResNet-18 as the backbone, pretrained by various KD methods over the Faster-RCNN. The numbers in **bold** and underline denote the best and the second-best results, respectively.

| Method | mAP | AP(Average Precision) | | | | | | | | | | | | | | | | | | | |
|---|---|---|---|---|---|---|---|---|---|---|---|---|---|---|---|---|---|---|---|---|---|
| | | aero | bike | bird | boat | bottle | bus | car | cat | chair | cow | table | dog | horse | mbike | person | plant | sheep | sofa | train | tv |
| Baseline | 76.2 | 79.0 | 83.4 | 77.5 | 63.4 | 65.0 | 80.3 | 85.5 | 86.7 | 59.4 | 81.7 | 69.1 | 84.2 | 84.2 | 81.3 | 83.7 | 48.3 | 80.0 | 73.6 | 82.6 | 74.6 |
| KD | 77.1 | 78.6 | 84.4 | 78.0 | **68.0** | 62.8 | 82.9 | 85.9 | **88.4** | 60.7 | 81.5 | 68.6 | 85.9 | 84.4 | 82.7 | 84.5 | 48.8 | 79.5 | 74.8 | **85.0** | 75.9 |
| CRD | 77.4 | 77.5 | **84.9** | 77.3 | 66.3 | 65.6 | 82.3 | **86.6** | 88.2 | 62.1 | 81.8 | 70.8 | 85.7 | 84.9 | 82.8 | 84.3 | 51.6 | **81.1** | 75.2 | 83.1 | 75.2 |
| SSKD | 77.6 | 78.6 | 84.2 | 78.2 | 66.2 | 63.3 | 82.8 | 86.1 | 87.3 | **63.4** | 84.4 | **71.8** | 84.5 | 85.1 | 83.1 | 83.9 | 51.9 | 80.1 | **77.2** | 84.1 | 76.0 |
| ours | **78.4** | **81.2** | 83.8 | **81.6** | 67.4 | **68.0** | **83.7** | 85.9 | 87.4 | 63.0 | 83.8 | 68.4 | **86.8** | **86.2** | **83.7** | **85.0** | **53.7** | **81.1** | 74.8 | 84.4 | **78.6** |

The comparison of detection performance toward average precision (AP) on individual classes and mean AP (mAP) is shown in Table. 14. Our method outperforms the original baseline by 2.24% mAP and the best-competing SSKD by 0.82% mAP. Moreover, our method can also achieve the best or second-best AP in most classes compared with other widely used KD methods. Notably, our method has achieved the best AP in 12 categories, with a significant improvement observed in the bird category, surpassing the second-best AP achieved by SSKD by 3.4%. Furthermore, we have attained the second-best AP in 4 categories, with an average difference of 0.5% from the best AP. These results verify that our method can guide a network to learn better feature representations for semantic recognition tasks.

## A.6 LIMITATIONS

Since our method needs to first train the teacher's SM module and obtain the relationship matrix of angle and length, it takes a longer time than the traditional KD method. For example, in a single GPU Tesla-V100 to the res32x4→res8x4 pair, the training time on the CIFAR-100 dataset for traditional KD is around 90min, whereas for MDR is around 470min.

In addition, compared with other relational distillation methods that only use the penultimate layer information for distillation, this method needs to use the middle layer information, so the teacher and the student need to have the same number of stages. Therefore, there is a constraint on the selection of distillable networks.

## A.7 VISUALIZATIONS

In this part, we present some visualizations to show that our MDR does bridge the teacher-student gap in the relation-level. In particular, we visualize the MSE loss of relational matrix between ResNet32×4 and ResNet8×4 in Fig. 5. We find that our MDR significantly improves the similarity of angle-wise, length-wise and distance-wise relational matrix between the student and the teacher.

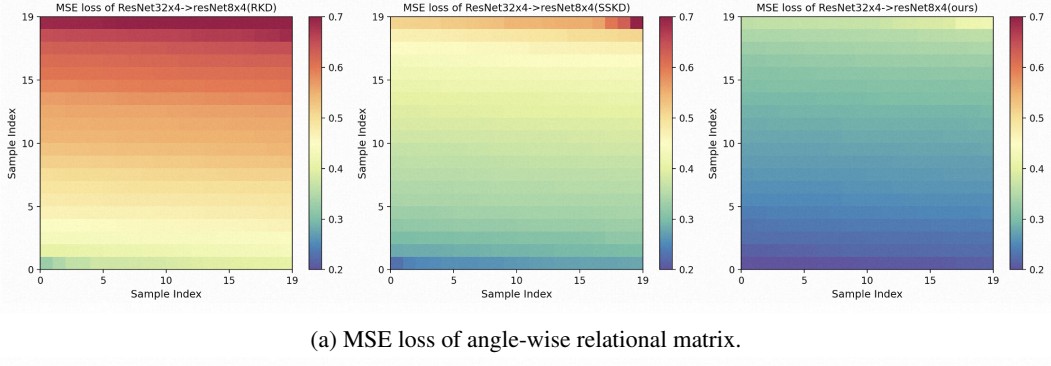

(a) MSE loss of angle-wise relational matrix.

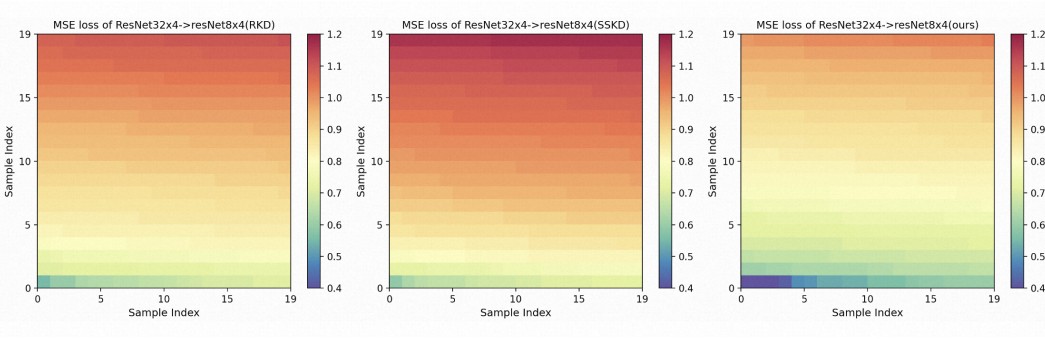

(b) MSE loss of length-wise relational matrix.

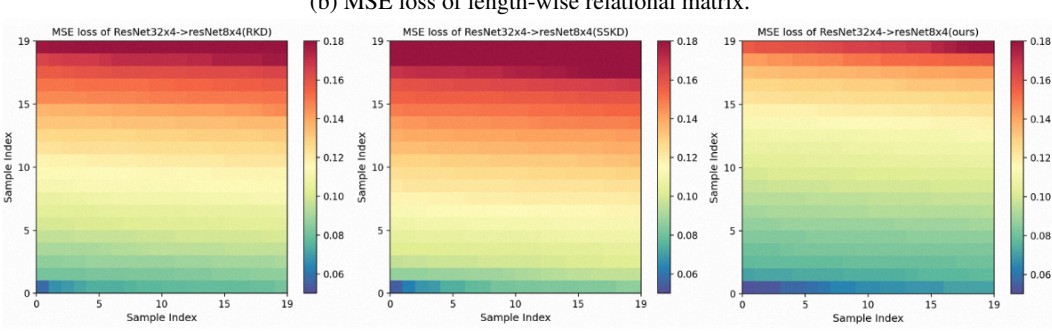

(c) MSE loss of distance-wise relational matrix.

Figure 5: **MSE loss of relational matrix between ResNet32×4 and ResNet8×4.** We visualize the MSE loss of angle-wise (top) ,length-wise (middle) and distance-wise (bottom) relational matrix between the models trained by RKD (left), the models trained by SSKD (middle), and the models trained by our MDR (right). The experiments are conducted on the sampled CIFAR-100 validation set (10,000 samples). We compute relational matrix with a batch size of 25 for the penultimate stage, so these are 400 values for each experiment. For better presentation, we rank these values and organize them as the heatmap representation. The smaller the value, the more similar the matrix are.

