# OpenReview forum: "Teaching wiser, Learning smarter: Multi-stage Decoupled Relational Knowledge Distillation with Adaptive Stage Selection"
_ICLR.cc/2024/Conference — Submitted to ICLR 2024_

### Official Review · Reviewer_zgd9 · 2023-10-30

**Soundness:** 3 good
**Presentation:** 2 fair
**Contribution:** 2 fair
**Rating:** 5
**Confidence:** 4

**Summary:**

This article proposes a method for relation-based knowledge distillation using contrastive learning. Its main contribution lies in utilizing multi-stage feature outputs and decoupling angle and distance information in the distillation process.

**Strengths:**

1. The article discusses the shortcomings found in previous works and introduces some innovative ideas.
2. Decouple the imformation of angle and length, which seems to be effective.
3. Using multiple layers of features for effective knowledge distillation in classification tasks is meaningful. While multi-layer features have shown significant improvements in detection tasks, in classification tasks, it is common to use only the last layer features. Exploring the rational use of multiple layers of features is worthy of investigation.

**Weaknesses:**

1. The effectiveness of the proposed method is evident on CIFAR-100, but there is limited comparison on ImageNet 1K, and the improvements seem insufficient. Please provide additional experiments on ImageNet.
2. Please include comparisons with latest methods，such as MGD, DIST and NKD.

   DIST: Knowledge distillation from a stronger teacher. NeurIPS.

   MGD: Masked Generative Distillation. ECCV

   NKD: From Knowledge Distillation to Self-Knowledge Distillation: A Unified Approach with Normalized Loss and Customized Soft Labels. ICCV
3. Please provide further explanations and performance comparisons regarding the ADSS module.
4. Some parts of the article employ unnecessarily complex language, which hinders readability.

**Questions:**

above

---

> ### Author Response · Authors · 2023-11-21
>
> Thanks for your constructive comments and recognition of our work. Responses to specific questions are provided below and will be modified in the revised manuscript.
> ### Q1:
> Additional experiments on ImageNet.
> ### A1:
> The data in Table 2 in the original paper refer to the experimental results of DKD. For a fair comparison, we reproduce SSKD and the most recent advanced methods ML-LD by using the same data augmentation MixUp both on ResNet34→ResNet18 and ResNet50→MobileNetV1 teacher-student pairs, the results have been updated to the revised version.
> The results show that our MDR improves the accuracy by 0.51% over SSKD for Top-1 accuracy on ResNet34→ResNet18 and 0.36% on ResNet50→MobileNetV1. From the current comparison methods, we have achieved SOTA results.
> Due to the time and resource limitation, we will reproduce all methods in Table 2 in the camera-ready version.
> ### Q2:
> Comparisons with latest methods, such as MGD, DIST and NKD.
> ### A2:
> We have reproduced all mentioned methods and the most recent advanced methods ML-LD on CIFAR100, the results have been updated to the revised manuscript.
> ### Q3:
> Further explanations and performance comparisons regarding the ADSS module.
> ### A3:
> We observed that transferring teacher's output to relational knowledge through contrastive learning improves student's accuracy. However, transferring the output of all stages simultaneously doesn't further enhance performance.  This suggests that multi-stage knowledge transfer may introduce redundant and unfavorable information. So we filter them and propose ADSS. To address this, we introduce ADSS, which summarizes knowledge at each stage and uses adaptive selection strategies for angle-wise and length-wise relationships decoupled by RDM.
> __For the angle-wise relationship:__ We propose relative numerical ranking. We calculate cosine similarity between positive samples and negative samples in the same batch at each stage, and rank them. We then compare rankings across stages, selecting stages with favorable information and discarding those with unfavorable information. Ablation experiments show that selecting one stage per sample often yields the best results.
> __For the length-wise relationships:__ We show (as seen in Table 11 in the revised pdf) that using the relative shortest distance as the criterion yields the best overall performance. This criterion shifts from cosine similarity between samples to the actual distance between them. We still gauge representation ability by ranking corresponding anchors and positive samples in the same batch, allowing us to select stages with favorable information. Our ablation experiments confirm that choosing one stage per sample often leads to optimal results.
> __Performance Comparison:__ We compared the information of all stages and the stage information filtered by ADSS on different network pairs, as shown in Table 12. We also conducted experiments on angle and length respectively. In order to compare the differences more intuitively, we changed the angle-related strategy while keeping the length-related one unchanged, and vice versa. The results show that whether it is angle or length, the distillation effect can be significantly improved after using ADSS.
> ### Q4:
> Readability improvement.
> ### A4:
> We will modify some expressions to improve readability in the revised version.

---

### Official Review · Reviewer_G2bH · 2023-11-01

**Soundness:** 3 good
**Presentation:** 3 good
**Contribution:** 3 good
**Rating:** 8
**Confidence:** 4

**Summary:**

This paper focuses on knowledge distillation (KD) where a (usually more compact) student model has to be trained using a (usually heavier) teacher model. They extend the ideas from relational knowledge distillation (RKD) in order to better utilize information from multiple intermediate stages of the models. They find that adding distance-wise relational information to contrastive learning based methods deteriorates the distillation performance. To rectify this, they propose Multi-stage Decoupled Relational KD (MDR) to decouple the distance-wise information into length-wise and angle-wise information. Intuitively, it seems to be easier to optimize the length and angle separately instead of implicitly through a single distance measure. They extensively evaluate the proposed method on KD benchmarks like CIFAR100 and ImageNet, as well as on few-shot learning, object detection, and transfer learning.

**Strengths:**

* The idea of decoupling the distance information into length and angle information is novel, interesting, and intuitive. I appreciate that it is a simple yet effective idea.

* The paper is fairly well-written and easy to follow.

* The analysis and ablation experiments are quite extensive and confirm that the proposed components are useful.

**Weaknesses:**

* Page 5: “RKD directly used distance information but ended up with unstable and degraded distillation outcomes.” Based on Table 1, there is only a 2% difference between RKD and the proposed method. This claim of unstable and degraded outcomes seems exaggerated without proper evidence.

* Page 6: “To preserve length-wise information while maintaining the representational ability of contrastive learning, we place a classifier behind each SM and directly use cross-entropy (CE) loss.” There are not many details given about this and it is not illustrated in Fig. 2.
    * If only CE loss is used, then how does it get contrastive representational ability?
    * The first contribution mentions that this work improves existing contrastive learning based KD frameworks, but the contrastive loss is removed here. This seems contradictory.
    * Also, there is no loss equation mentioned for SM training and no information on the loss weighting hyperparameters involved in SM training.
    * The ablation study for SM training (Fig. 3c) only compares the SMP accuracy but not the downstream distillation performance. Please also show the distillation performance since we need to have a proper comparison to understand the benefits of changes to SM training.

* Fig. 3b: It is unclear what ${\mathcal{L}}\_{len}$ represents because the text mentions it is distance-wise loss but it was defined as length-wise loss. Also it is unclear what it means by adding RDM. Because RDM was defined as having separate losses $\mathcal{L}\_{len}$ and $\mathcal{L}\_{ang}$ which is the second bar in the figure.

* Fig. 3b: Another baseline is using the two decoupled losses and a third loss being the distance-wise loss. This needs to be checked in case having all three losses may further improve the optimization.

* There are different hyperparameters for different datasets (as per Page 13) without any information on how they are chosen. Due to the large number of hyperparameters, the method may be difficult to tune.
    * Also, there are no sensitivity analyses for each of these hyperparameters.
    * As mentioned before, there is no information on the hyperparameters involved in the CE losses for the self-supervised modules.

* Fig. 5: Please also add the visualization of distance-wise relational matrix, so we can see if that loss is also lower for the proposed MDR. It would also support the idea that decoupled optimization improves the solution to the overall goal optimization problem.

**Questions:**

* Please see the weaknesses section.

* Minor comments
    * Fig. 2: Please mention what the red cross indicates in the legend.
    * Eq. 5: use \exp instead of exp.
    * Table 1: add a column for “average” to make it easier for readers to understand the average improvements across all architectures.
    * Fig. 3: x-axes of (a) and (b) have typos: “shffule” → “shuffle”.
    * Fig. 3c: Indicate what 1st, 2nd, 3rd means in the text (I believe it is the layer number or stage number of the SM module). Also it is unclear if the plot is for the teacher SM or the student SM.
    * Sec. A.4: typo in the title “Ablition” → “Ablation”.

---

> ### Author Response · Authors · 2023-11-21
>
> Thanks for your constructive comments and recognition of our work. Responses to specific questions are provided below and will be modified in the revised manuscript.
> ### Q1:
> About the wording of related works, spelling errors and additional visualization for Fig.2 and Fig.3.
> ### A1:
> All modifications and additions have been updated and highlighted in our revised manuscript.
> ### Q2:
> Details about SM designs and training strategy.
> ### A2:
> Due to space limitations, we provide more details on SM in Appendix A.2 of the revised manuscript. Here we highlight the key points.
> The functions of SM include:
> + Enhancing the comparative learning representation ability of the teacher network to better guide students in learning relationships;
> + Maintaining angle-wise and length-wise information between samples;
> + Unifying the representational spatial dimensions of teachers and students.
>
> In the distillation process, we first train the teacher's SMs separately. To retain the length-wise information, we modify the SM training method by directly connecting it to the classifier and train with the original task while keeping the teacher's backbone frozen. We then connect the student's SM after each stage to transfer knowledge, where both the student's SM and backbone are involved in distillation training.
> Regarding the use of CE loss, it actually complements our contrastive learning-based KD framework due to the following reason. We modified the teacher's SM training method, which, experimentally, improved both length-wise information retention and contrastive representational ability compared to traditional training methods. Additionally, our framework continues to utilize positive and negative sample relationship extraction based on contrastive learning for angle-wise relationship extraction.
> The reason for using CE loss to enhance contrastive representational ability lies in its ability to make positive samples close to anchors and negative samples distant in hyperplane space. For classification tasks, this aligns with the anchor and positive sample being of the same category, resulting in similar one-hot encoded classification labels. Conversely, the anchor and negative sample, from different categories, exhibit vertical separation in one-hot encoding, making CE loss suitable for learning contrastive representations.
> ### Q3:
> Information on how to choose hyper-parameters for different datasets, and sensitivity analysis for each of these hyper-parameters.
> ### A3:
> For the weights of $L_{cls}$ and $L_{kd}$, we followed the weight of SSKD. The weight of $L_{len}$ has different values on different datasets. We want to make each loss component reach similar order of magnitude as much as possible to balance the role of each component. Since the loss of KL divergence itself is too small, we give a larger weight. We also added the sensitivity analysis for $L_{ang}$ and $L_{len}$ in the Appendix A.4 of the revised manuscript.

---

> > ### Comment · Reviewer_G2bH · 2023-11-23
> > **Response to authors**
> >
> > I thank the authors for their efforts during the discussion period. Most of my concerns have been addressed, so I upgrade my score to 8: accept, good paper since the strengths outweigh the weaknesses.

---

### Official Review · Reviewer_qoFL · 2023-11-01

**Soundness:** 2 fair
**Presentation:** 3 good
**Contribution:** 2 fair
**Rating:** 6
**Confidence:** 5

**Summary:**

This paper proposes a Multi-stage Decoupled Relational (MDR) knowledge distillation framework, which selects the most suitable stages for each sample based on the relational representation capability of both angle-wise and distance-wise relational information. It decouples angle-wise and distance-wise information to enhance the  transferring efficiency and distillation quality. The experiments show that the proposed method achieves SOTA performance on the public datasets.

**Strengths:**

The organization of this manuscript is satisfying and the proposed MDR achieves SOTA performance.

**Weaknesses:**

1. The relational decouple module (RDM) is not very novel, as the SSKD has already introduced contrastive prediction, the RKD also employs Angle-wise distillation loss for distillation.
2. The authors should compare the MDR with the most recent advanced methods, e.g. CAT-KD [1] and ML-LD [2].
[1] Ziyao Guo, Haonan Yan, Hui Li, Xiaodong Lin: Class Attention Transfer Based Knowledge Distillation. CVPR 2023: 11868-11877
[2] Ying Jin, Jiaqi Wang, Dahua Lin: Multi-Level Logit Distillation. CVPR 2023: 24276-24285

**Questions:**

1. What’s major insight of the RDM modules? the decoupling technique is not very novel in knowledge distillation task.
2. In table 1, in some cases the student network achieves higher performance than the teacher network, can you explain such phenomenon?

---

> ### Author Response · Authors · 2023-11-21
>
> Thanks for your constructive comments and recognition of our work. Responses to specific questions are provided below and will be modified in the revised manuscript.
> ### Q1:
> The major insight and the novelty of the relational decouple module (RDM).
> ### A1:
> The key idea of RDM is to separate sample relationships into angle-wise and length-wise aspects, resolving conflicts that arise from simultaneously supervising both angle-wise and distance-wise relationships, which can reduce the effectiveness of distillation. In RKD, there's a loss design involving both angle-wise and distance-wise relationships, but using both concurrently is often less effective than using them independently. We believe this is due to information coupling when learning angle-wise and distance-wise information simultaneously. In hyperplane space, distance between samples is influenced by both angle and length, creating conflicts when directly applying distance-wise loss to reduce sample distance. To address this, we propose a decoupling module to supervise both aspects simultaneously without information conflicts.
> Regarding novelty, while the concept of decoupling has appeared in other KD methods, such as DKD, our unique contribution lies in decoupling relationships between samples, a novel approach introduced for the first time in relation-based knowledge distillation methods.
> ### Q2:
> Comparison of MDR with the most recent advanced methods.
> ### A2:
> We reproduce ML-LD, the most accurate method among the recent advanced methods with the same data augmentation and classic KD, and compare it with MDR on CIFAR100 and ImageNet datasets. Our method can still achieve SOTA accuracy.
> The experimental results are shown in Tables 1 and 10 in the revised manuscript.
> ### Q3:
> Reasons why student networks perform better than teacher networks in some cases in table 1.
> ### A3:
> It can be explained from the following aspects:
> + We use the ADSS module to select the transferred information. For each sample, only the stage with the strongest representation ability is selected for knowledge transmission, thus filtering out some unfavorable information from teacher;
> + We use the MixUp data angmentation and propose a new SM training method to enhance relationship representation capabilities of the teacher model;
> + While doing knowledge distillation, we will also use the loss of the original task $L_{cls}$, and the CIFAR100 dataset is less difficult. Although a smaller model holds less knowledge, it has a smaller probability of overfitting than a larger model.

---

### Official Review · Reviewer_WUNU · 2023-11-01

**Soundness:** 2 fair
**Presentation:** 3 good
**Contribution:** 2 fair
**Rating:** 5
**Confidence:** 4

**Summary:**

This paper attempts to improve knowledge distillation based on contrastive learning. Specifically, the authors introduce two aspects of improvements. The first is applying contrastive feature distillation over intermediate layers facilitated with a so called adaptive layer-stage selection. The second is using two functions, namely cosine similarity and L2-normed distance, measure relational feature distance between student and teacher models. Experimental validation is conducted on image classification (with CIFAR-100 and ImageNet datasets) and object detection (with PASCAL VOC dataset) tasks. Besides traditional supervised image classification, transfer learning and few-shot learning are also considered in experiments.

**Strengths:**

+ The paper is well written in most parts.

+ The proposed method is simple and straightforward.

+ Comparative experiments are conducted on image classification (with CIFAR-100 and ImageNet datasets) and object detection (with PASCAL VOC dataset) tasks.

+ The proposed method shows competitive performance.

**Weaknesses:**

- The method and presentation.

This paper improves contrastive knowledge distillation from the perspective of leveraging relation features at multiple intermediate layers/stages instead of one single layer (typically at penultimate layer). The proposed method includes two improvements, both of which are straightforward. First, applying contrastive knowledge distillation to the teacher-student layer pair which has the highest similarity among a set of candidate layer pairs. Second, separately using cosine similarity and L2-normed distance to invoke two types of contrastive knowledge distillation. Although the authors call these two simple improvements as Adaptive Stage Selection and Relation Decouple Module, they are hand-crafted but not "adaptive" and "decoupled". Moreover, I have not seen any theoretical insights for the design/formulation of them. Also, why and how they work are not clear to some degree.

Besides, multi-stage designs are widely used in many existing knowledge distillation methods. Discussion of this line research is missing.

- The limitations.

The authors did not discuss the limitations of the proposed method.

- The experiments.

To comparative experiments on CIAFR-100 (Table 1), did the authors run the reference methods by themselves? Why the results for these methods are usually lower than the formal results reported in the original papers, e.g., SSKD which is closely related to the proposed method.

To comparative experiments in Table 1, Table 2 and Table 9 (in the Appendix), it seems that the results of the proposed method are obtained by using logits based KD at the head. I would like to see a more fair comparison, either using logits based KD to all methods or removing it to all methods.

Furthermore, the proposed method adopts mixup in training. In this context, the baseline with mixup should be reported and used as more proper baseline. And with mixup, the performance of the reference methods would be also improved.

From Table 6, the proposed method performs best with one single stage setting. This is weird to a large degree. Why?

**Questions:**

Please refer to my detailed comments in "Weaknesses" for details.

---

> ### Author Response · Authors · 2023-11-21
>
> Thanks for your constructive comments and recognition of our work. Responses to specific questions are provided below and will be modified in the revised manuscript.
> ### Q1:
> The naming and the theoretical insights of the proposed techniques.
> ### A1:
> The adaptivity in ADSS comes from the fact that suitable stages are selected dynamically for distillation based on different samples; hence we call it adaptive stage selection. We also convert the distance-wise relationship between samples into angle-wise and length-wise information. This decoupling prevents the interference in scenarios where conventional distance and angle supervision conflict with each other.
> __Insights to ADSS:__ we found that hyperplane representations vary significantly across stages, with cosine similarities increasing in deeper levels. Relying solely on absolute cosine similarity for distillation tends to favor the final stage. However,early stages also hold valuable information as shown in Fig. 1a. Hence, we introduced ADSS, which evaluates representational ability by comparing both the similarity of positive samples to the anchor and the distance of negative samples from the anchor. The similarity ranking of the anchor with positive samples is used as the final quantitative result. Regarding distance, we obtained the optimal distance criterion through extensive experimentation.
> __Insights to RDM:__ the decoupling of the distance-wise information is to avoid conflicts when learning the relationship matrix, e.g., the distance between samples increases whereas the angle difference reduces. The decoupling ensures that length-wise supervision does not influence angle changes, leading to a more representational relationship matrices without interference.
> ### Q2:
> Discussion of the multi-stage designs in existing knowledge distillation methods.
> ### A2:
> Existing knowledge distillation methods can be broadly divided into response-based, feature-based and relation-based. Using multi-stage information has become the prevailing approach for feature-based method, such as ReviewKD. Response-based methods using multi-stage information have also emerged, such as HSAKD. To the best of our knowledge, our work is the first multi-stage relation-based knowledge distillation method. We propose ADSS to filter useless and redundant information and RDM to resolve relationship conflicts between samples. More details are updated in the Related Work.
> ### Q3:
> The limitations of the proposed method.
> ### A3:
> + Compared with traditional KD, our method takes more time and resource to train.
> + Our method requires the number of model stages for both teachers and students.
>
> More details can be found in Appendix A.6 in the revised manuscript.
> ### Q4:
> About the data and comparisons in Tables 1, 2 and 9.
> ### A4:
> We did reproduce the data by ourselves in Table 1, as such there is a slight difference in accuracy from the original paper. The difference in accuracy remains even after we re-did the reproduction work.  Other articles citing SSKD (such as PACKD, HASKD) also had the similar accuracy difference. We believe that the authors fine-tuned the hyper-parameters of different networks.
> Also, we have reproduced all the mentioned methods and the most advanced method ML-LD by using MixUp and Logits KD on CIFAR100. Regarding ImageNet, we only reproduced the experiments of SSKD and ML-LD due to time and resource constraints.  Beacuse of the same data augmentation and logits KD, the improvement of our method compared to other methods has been reduced, but our method can still achieve SOTA accuracy. All results have been updated in the revised manuscript.
> ### Q5:
> The reason that MDR performs best with one single stage setting.
> ### A5:
> We found that multi-stage relationship information does not further improve accuracy (as shown in Fig. 1a.) Therefore, we introduced ADSS to eliminate redundant information. Extensive experiments with various networks revealed that using only one stage can achieve the best results. The reason is that unlike feature maps, multi-stage relationship information lacks potential progressive relationship, and redundant information may hinder distillation effectiveness. For different samples, ADSS adaptively selects  appropriate stage for distillation based on relative representation capabilities, so it is still a multi-stage distillation architecture.

---

### Meta-Review · Area_Chair_vqce · 2023-12-10

**Metareview:**

This paper endeavors to improve knowledge distillation through the lens of contrastive learning. It has been thoroughly reviewed by four experts. The reviews received are mixed, reflecting a nuanced view of the paper's contributions and limitations. The reviewers commend the paper for its generally well-structured and clear presentation.

However, despite these strengths, there are significant concerns that need to be addressed. A key issue raised pertains to the empirical evaluation of the proposed method. Specifically, the reviewers note that the comparisons on ImageNet-1K are limited, and the improvements demonstrated are not substantial enough to robustly support the claims made in the paper.

Given these considerations, the decision at this stage is to not recommend the paper for acceptance. This outcome, however, should not discourage the authors. We encourage the authors to take these comments into account, particularly focusing on expanding and strengthening the empirical comparisons.

**Justification For Why Not Higher Score:**

There are limited comparisons on ImageNet-1K, and the improvements are insufficient to support the claims.

**Justification For Why Not Lower Score:**

N/A

---

### Decision · Program_Chairs · 2024-01-16

Reject